# Evolution of Animal South American RVA Told by the NSP4 Gene E12 Genotype

**DOI:** 10.3390/v14112506

**Published:** 2022-11-12

**Authors:** Samuel Orlando Miño, Alejandra Badaracco, Enrique Louge Uriarte, Max Ciarlet, Viviana Parreño

**Affiliations:** 1Instituto Nacional de Tecnología Agropecuaria INTA, EEA Cerro Azul, Ruta Nacional 14, km 836 (CP3313), Cerro Azul 3313, Argentina; 2Instituto Nacional de Tecnología Agropecuaria INTA, EEA Montecarlo, Av. El Libertador N° 2472 (CP3384), Montecarlo 3384, Argentina; 3Instituto Nacional de Tecnología Agropecuaria INTA, EEA Balcarce, Ruta 226 km 73.5 (CP7620), Balcarce 7620, Argentina; 4Clinical Development, Icosavax, Seattle, WA 98101, USA; 5Instituto Nacional de Tecnología Agropecuaria INTA, CICVyA, INTA Castelar, De las Cabañas y De los Reseros s/n (CP1686), Hurlingham 1686, Argentina; 6Instituto de Virología e Innovaciones Tecnológicas (IVIT), CICVyA, INTA Castelar, De las Cabañas y De los Reseros s/n (CP1686), Hurlingham 1686, Argentina

**Keywords:** genome reassortment, phylogeny, evolution in segmented viruses, phylodynamic

## Abstract

Rotavirus A (RVA) possesses a genome of 11 double-stranded (ds) RNA segments, and each segment encodes one protein, with the exception of segment 11. NSP4 is a non-structural multifunctional protein encoded by segment 10 that defines the E-genotype. From the 31 E-genotypes described, genotype E12 has been described in Argentina, Uruguay, Paraguay, and Brazil in RVA strains infecting different animal species and humans. In this work, we studied the evolutionary relationships of RVA strains carrying the E12 genotype in South America using phylogenetic and phylodynamic approaches. We found that the E12 genotype has a South American origin, with a guanaco (*Lama guanicoe*) strain as natural host. Interestingly, all the other reported RVA strains carrying the E12 genotype in equine, bovine, caprine, and human strains are related to RVA strains of camelid origin. The evolutionary path and genetic footprint of the E12 genotype were reconstructed starting with the introduction of non-native livestock species into the American continent with the Spanish conquest in the 16th century. The imported animal species were in close contact with South American camelids, and the offspring were exposed to the native RVA strains brought from Europe and the new RVA circulating in guanacos, resulting in the emergence of new RVA strains in the current lineages’ strongly species-specific adaption. In conclusion, we proposed the NSP4 E12 genotype as a genetic geographic marker in the RVA strains circulating in different animal species in South America.

## 1. Introduction

Rotavirus A (RVA) belongs to the Reoviridae family (genus Rotavirus, Rotavirus species A). The rotavirus particle is composed of three concentric capsid layers, with a genome of 11 double-stranded (ds) RNA segments, and each segment encodes one protein only with the exception of segment 11. The segments 1–6 encode the virion proteins (VP), while the remaining segments encode the non-structural proteins (NSPs) [1]. The rotavirus NSPs then coordinate various stages of genome replication and viral assembly by adapting and modifying the cellular machinery, which leads to productive release of mature particles through cell lysis [2].

RVA strains have been classified since 2008 based on the complete genomic constellation describing the genotype of all eleven segments [3]. With the advent of sequencing, all new and novel detected RVA strains, and the reference strains were fully sequenced. With the current classification scheme, the number of NSP4 genotypes (E-types) increased from 12 to 31 at the present time [3]. NSP4 is one of the better-characterized non-structural multifunctional proteins of RVA strains. Encoded by gene segment 10, NSP4 is a 175 amino acid transmembrane protein, essential for RVA replication, transcription, and morphogenesis. In addition, NSP4 has been found in dimeric, tetrameric, pentameric and higher-ordered multimeric structures, showing a highly conserved region from amino acids 95–135. This region encompasses the enterotoxigenic peptide 114–135 that elicits diarrhea in neonatal mice [2].

The E12 genotype of NSP4 has been reported in RVA-detected cows from Argentina [4] and Uruguay [5]. Moreover, RVA strains carrying and E12 NSP4 have been detected in Argentinian horses [6], goats [7] and guanacos [4]. Furthermore, the E12 NSP4 genotype has been found in RVA strains infecting humans in Paraguay [8] and Brazil [9]. Interestingly, the E12 has only been detected in RVA circulating in South American countries.

The segmented genome of RVA allows the exchange of genome segments during co-infection through a process called gene reassortment. Specifically, when two or more viruses infect a single host cell, they can package each other’s genome segments into a nascent virion, thereby producing hybrid progeny [10]. The genetic exchange requires the conservation of assortment signals and preservation of the RNA–RNA and/or RNA–protein interactions that mediate genome packaging and replication. For this reason, reassortant viruses selectively emerge at appreciable levels in the viral population with a genomic composition that confers at least some modest advantage to viral fitness [10]. Reassortment is also observed as a host restriction factor, a mechanism described among RVA strains [11]. Taken together, the segmented nature and the interspecies transmission of rotavirus shows the zoonotic relevance and the reason why this disease should be faced from the One Health approach [12].

In this work, we studied the evolutionary relationships of RVA strains carrying the E12 NSP4 genotype detected in domestic and wild animal species, as well in humans in South America by phylogenetic and phylodynamic analyses of the gene encoding NSP4.

## 2. Materials and Methods

Dataset and phylogenetic analysis: Two datasets were constructed, one including all the NSP4 reported genotypes (All NSP4) in order to study the relationships of E12 with respect to the others, and one with all E12-genotypes only (NSP4-E12) to conduct a Bayesian analysis.

NSP4 dataset: Sequences from all NSP4 genotypes were downloaded from the Genbank database on 10 June 2021. Into the virus variation database, the search was defined as: (a) sequence type: nucleotide; (b) specie: rotavirus A; (c) host: any; (d) region: any; (e) segment: NSP4; (f) isolation source: any (https://www.ncbi.nlm.nih.gov/genomes/VirusVariation/Database/nph-select.cgi, accessed on 16 October 2022). The sequences were aligned in MAFFT [13] and trimmed to the size of the open reading frame (ORF) using Aliview [14]. We defined 95% of the open reading frame as the cut-off to include a strain sequence into the analyses. All strain sequences under cut-off were excluded. A total of 5380 sequences from human and animal species were used to construct the NSP4 dataset. The molecular evolutionary models were inferred according to the Bayesian information criterion (BIC) statistics obtained with ModelFinder [15], and the maximum-likelihood (ML) phylogenetic trees were inferred using IQ-Tree 1.5.5 [16] with Ultrafast bootstrap [17] and the Shimodaira–Hasegawa-like approximate Likelihood Ratio Test [18] as branch support (UBoot = 10,000 and aLRT = 10,000 replicates). Trees were plotted using FigTree (available at https://github.com/rambaut/figtree, accessed on 16 October 2022).

The NSP4-E12 dataset was constructed using sequences from our lab as well as all NSP4 E-12 genotypes from the Genbank database (*n* = 98) (Table 1) and Badaracco A.’s [19] and Miño S.’s [20] PhD theses (accession numbers OP676137-OP676207). E2-genotype strains were included as outgroups (*n* = 8). The final NSP4-E12 dataset was 106 taxa of 525 nucleotides each. Sequences were aligned using Muscle v7.407 [21], implemented in Aliview and trimmed to the size of the ORF.

Prior to Bayesian or maximum likelihood phylogenetic analyses, the information contained in the dataset was assessed. The quality analysis showed a base frequency (pi) of pi(A) = 0.39, pi(C) = 0.15, pi(G) = 0.20, pi(T) = 0.26. Besides, the mutation rates were A-C = 1.00, A-G = 28.28, A-T = 1.00, C-G = 1.00, C-T = 28.28, and G-T = 1.00. The estimated evolutionary model was Hasegawa–Kishino–Yano with 4 gamma categories (HKY+G4) [22]. In addition, 60% of invariant sites (317/525) were also estimated. The recombination was evaluated by the PHI test [23] implemented in SplitsTree4 [24], and non-recombination events were detected. The presence of a phylogenetic signal was assessed by a likelihood mapping method [25], implemented in IQ-Tree. The presence of a temporal signal was examined by root-to-tip regression using TempEst v1.5.1 software (University of Edinburgh, Edinburgh, UK) [26]. 

A Bayesian coalescent approach was used to estimate the phylogenetic relationship, divergence times, and the population dynamics of NSP4. The analysis was carried out in BEAST v1.8.4 package [27], implemented in the CIPRES server [28]. The NSP4 dataset was analyzed, and it was introduced as a partition with the corresponding molecular evolutionary model. All the analyses were carried out by setting up flexible models, i.e., a Bayesian skyline plot (BSP) demographic model and an uncorrelated lognormal (UCLN) molecular clock, and the temporal calibration was based on the tip dates. Markov chain Monte Carlo (MCMC) sampling was performed in duplicates, and samples were examined with Tracer v1.6 (available at https://beast.community/analysing_beast_output, accessed on 16 October 2022) to evaluate the convergence of parameters (effective sample size (ESS) of ≥200, acceptable mixing without tendencies in traces, with a burn-in of 10%). The maximum clade credibility (MCC) tree was summarized using Tree Annotator v1.8.4 and visualized with FigTree. The posterior probability upper to 90% and the year of the most recent common ancestor (MRCA) are shown on the tree nodes.

## 3. Results

Two percent (98/5380) of the strains selected to conduct the present study carried an E12 NSP4 genotype. The E12 genotype appeared as a monophyletic group with a 99% support value in both UFB and aLRT tests (Figure 1a).

The Bayesian analysis showed that the evolution rate (median mutation rate) for the E12 genotype is 4.93 × 10^−4^ (CI: 3.2–7.7 × 10^−4^) and the coefficient of variation was 0.338. The time of the most recent common ancestor (TMRCA) for the E12 genotype was calculated to be 1640 (credible interval 95%= 1435–1815). Additionally, there are two divergent strains at the root of the E12 genotype, one found in a guanaco with diarrhea from Río Negro (Patagonia, Argentina), RVA/Guanaco-wt/ARG/Río_Negro/1998/G8P [1] strain; and the other from a rare RVA strain, RVA/Cow-wt/URY/LVMS3024/2016/G24P [29], from an asymptomatic calf in Uruguay. A strong geographic structure was observed in the tree because all E12 strains were detected in South American countries. Within the E12 genotype, three major groups or lineages are observed according to different animal species: (i) horse lineage (pp = 100%) which included equine strains only; (ii) cow-G6 lineage (pp = 100%) which included bovine G6 strains exclusively, except for the RVA/Cow-wt/ARG/B2659_B_BA/2004/G10P [11] strain; (iii) inter specie lineage (pp = 100%), which contained bovine G10P [11], caprine G8P [1], guanaco G8P [1], human G8P [1] and human G10P [9] strains (Figure 1b). In addition, the population dynamic showed three behaviors through time. First, it showed a constant population size (a); a size population increase (b), and finally a constant population size again (c). Interestingly, these three zones were in concordance with different historical times of the Spanish conquest of South America, (from approximately 1492 to 1800), the establishment of free countries (since approximately 1800,) and current times (Figure 1c).

## 4. Discussion

Viral evolution is a phenomenon highly studied in several viruses, but most of the time viral evolution is studied in a short period of time due the nature of viruses [30]. However, segmented viruses as RVAs possess not only point mutations and intragenic recombination mechanisms, but also possess reassortment as an important mechanism to drive diversity [10]. In this work, we studied the evolutionary relationships of animals and RVA described in South America by the analysis of the NSP4 gene. Interestingly, a strong association between the introduction of horses and cows into the South American continent coincided with the evolution of the NSP4 gene in the continent. As a result, the RVA strains detected in South American cattle, thoroughbred horses, goats and wild guanaco (a native South American camelid) carry a NSP4 E12 genotype that has been only detected in this continent, and thus serves as a geographical genetic marker.

The RVA NSP4 E12 genotype was described for the first time in a South American animal in 2009 [4]. We studied all NSP4 sequences available in the rotavirus database on NCBI and the phylogenetic tree showed that all E12 genotype strains (2%) clustered together as a monophyletic highly supported branch. This phenomenon was also previously observed in the VP7 gene where the equine G14 genotype appears as a branch into the G3 genotype group at the nucleotide level [6]. However, the E12 genotype branch appears as a monophyletic group with strong support values (100%).

Viruses within similar hosts, such as hosts that reside in the same geographic region, are expected to be more closely related genetically if transmission occurs more commonly between them [31]. This phenomenon is observed as a concordance between geography and genetic results. A strong geographic structure was observed where all South American strains are only included within the E12 NSP4 branch and no E12 strains clustered outside the E12 NSP4 branch [31].

Bayesian analysis showed that the TMRCA for E12 genotype (1640) is in concordance with the introduction of the first cows and horses into South America from Europe (1492 onwards) [32,33]. In addition, in the RVA strain, the root of the E12 branch was determined to be that of the RVA/Guanco-wt/ARG/Río_Negro/1998/G8P [1] strain, as an ancestral strain from a wild young guanaco (chulengo) with diarrhea that was obtained during the capture of wild specimens in the Rio Negro province, Patagonia region. Interestingly, the guanaco RVA strain carried the same genome constellation as the other RVA guanaco strain obtained from a captured animal in the Chubut province, also in Patagonia [4]. Finally, the reassortant strain RVA/Cow-wt/URY/LVMS3024/2016/G24P [29] identified recently in calves from Uruguay [5] was distantly related to other E12 strains. A reassortant and/or point mutations event could be the reason why this rare strain clustered in a different branch than the other strains. 

Moreover, the phylogenetic tree showed three well-defined branches, one including equine strains only, another grouping mainly bovine G6 strains, and the last one composed of RVA from several animal species including human strains, suggesting a strong adaptation of the NSP4 gene to the different species [31]. An observation was proposed by others who explained that the viral proteins need to interact properly with each other in order to reach the best viral fitness for a certain host [10]. Therefore, the equine strains acquired characteristics that enhance fitness to infect horses, and the bovine strains to infect cows, respectively. These findings are consistent and supported by the observations that NSP4 gene or lineages tent to be a highly specie-specific adapted [29].

RVA infects young animals and, if the infection is cleared, the animal will have complete or partial immunity for the rest of its lifespan [19]. Because the animals possess an immature immune system, the pressure of selection may be diminished [34]. The low selection pressure is observed as a tree topology where the external branches are shorter relative to branches on the interior of the tree. In addition, the constant population size shows an infection with no selective pressures [31].

We proposed a hypothesis for the origin of the gene NSP4 E12 combining history and molecular evolution. Pedro de Mendoza arrived at the Río de la Plata in 1536 and founded the city of Buenos Aires. A few years later, Santiago de Valdivia founded “Santiago del Nuevo Extremo”, nowadays Santiago de Chile (Figure 2a) [32]. Originally, the wild biggest ruminant in the Río de la Plata was the guanaco (*Lama guanicoe*), which was present from the south of Argentina and Chile to Peru (https://camelid.org/, accessed on 16 October 2022). There is a report that shows Pedro de Mendoza riding a guanaco [35] and, as Pedro died shortly (1537) after founding Buenos Aires [36], one could assume that he was in contact with wild guanacos in Buenos Aires (Figure 2b). Four years later (1540), the Spanish left the city, pushed away by the natives of the lands (the Querandíes) leaving the horses and cows that they brought as free animals [37,38]. Without any natural predators in the Pampas, they bred freely, sharing the same environment with the guanacos (Figure 2a) [39]. Considering that the horses and cows and guanacos could be carriers of RVA strains together with the possibility of interspecies transmission, the horses and cows could have been infected by both old-world and new-world rotavirus coming from guanacos. Therefore, because RVA must carry only one copy of each gene only, the substitution of one segment by another could have occurred [10]. Thus, the E12 genotype originally from an autochthonous RVA was introduced by gene replacement (Figure 2c). Its hypothesis could explain why the E12 genotype is present on RVA circulating in South America only. During the last 20 years, our group has been studying the epidemiology and pathogenicity of rotaviruses in several animal species, resulting in several publications, including those of Matthijnssens et al., 2009 [4], Garaicoechea et al., 2011 [6], Louge Uriarte et al., 2014 [7], Miño et al., 2017 [20] and others. Our findings have revealed that rotavirus strains from different animal species always carried the NSP4 E12 genotype, while sharing the typical backbone. The other 10 genes possess genotypes that have been reported around the world, and they do not show a geographic pattern like the NSP4 E12 genotype does.

For many years, we did not find an explanation for this observation, until now, when we performed a Bayesian analysis that allowed us to determine a plausible date, rather remarkably, with the date of the introduction of horses and cows in South America by Europeans in the 16th century. Therefore, we initiated a detailed investigation in an attempt to understand the history of cattle and wild animals in South America. At the same time, additional investigations were being carried out, all of which confirmed our hypothesis. Initially, we did not consider finding a geographic marker in an ordinary find. However, a review of the historical literature provided additional evidence that supported to our hypothesis, which precipitated us to perform a comprehensive analysis of the current investigation as a standalone because of the importance it has. Hence the current article with the hypothesis that we present herein.

## 5. Conclusions

In conclusion, we found that the NSP4 E12 genotype phylogeny possesses a strong geographic structure with a common ancestor. We hypothesized a possible origin of the E12 genotype, and we support the concept that the NSP4 E12 genotype is a geographic marker for animal RVA strains from South America.

## Figures and Tables

**Figure 1 viruses-14-02506-f001:**
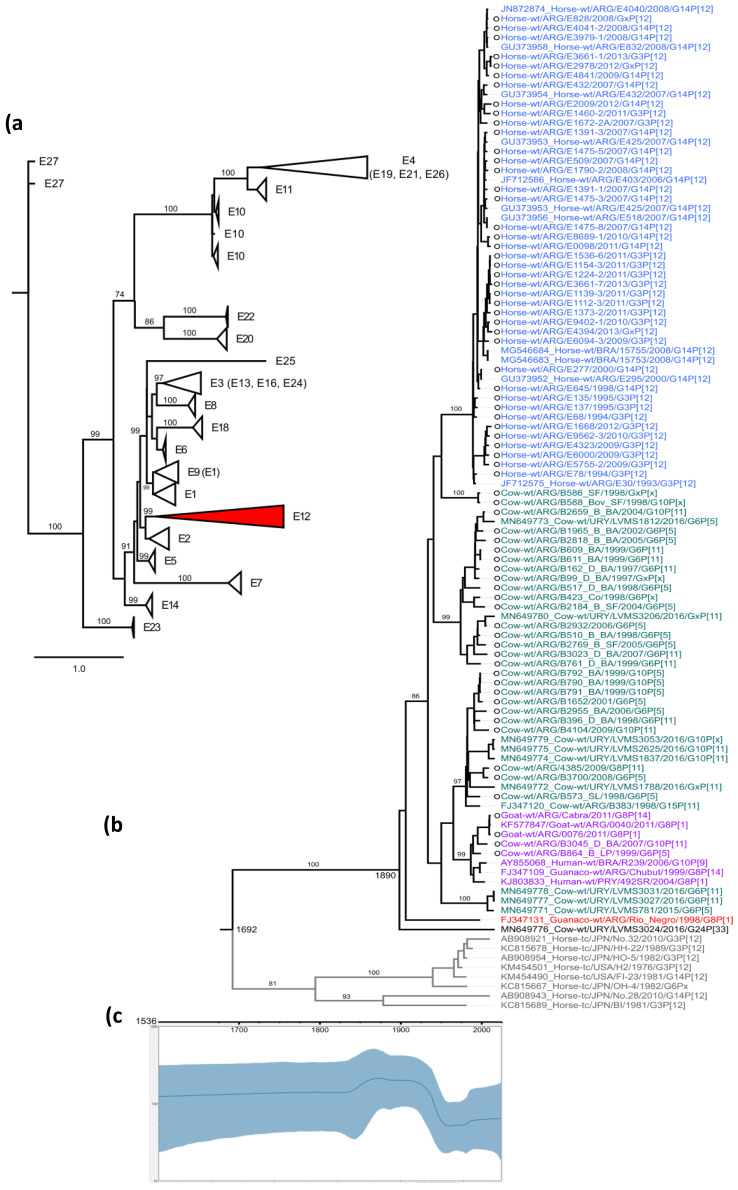
(**a**) NSP4 maximum likelihood phylogenetic tree showing the genotype clustering. Genotypes that clustered within a branch are shown in parentheses. The E12 genotype is colored in red. (**b**) Phylodynamic NSP4 E12 genotype. Branches are colored according to their host (equine, blue; bovine, green; guanaco, red; interspecies transmission, purple; out-group, gray). Sequences obtained in this study are shown with a circle on strain name. (**c**) Population dynamics of RVA NSP4 E12 genes. The effective numbers of infections through time (Neτ) obtained from the analysis of NSP4 E12 genes, under the UCLN-BSP models. The solid lines denote median values, while blue shadows denote the 95% highest posterior density (HPD) values.

**Figure 2 viruses-14-02506-f002:**
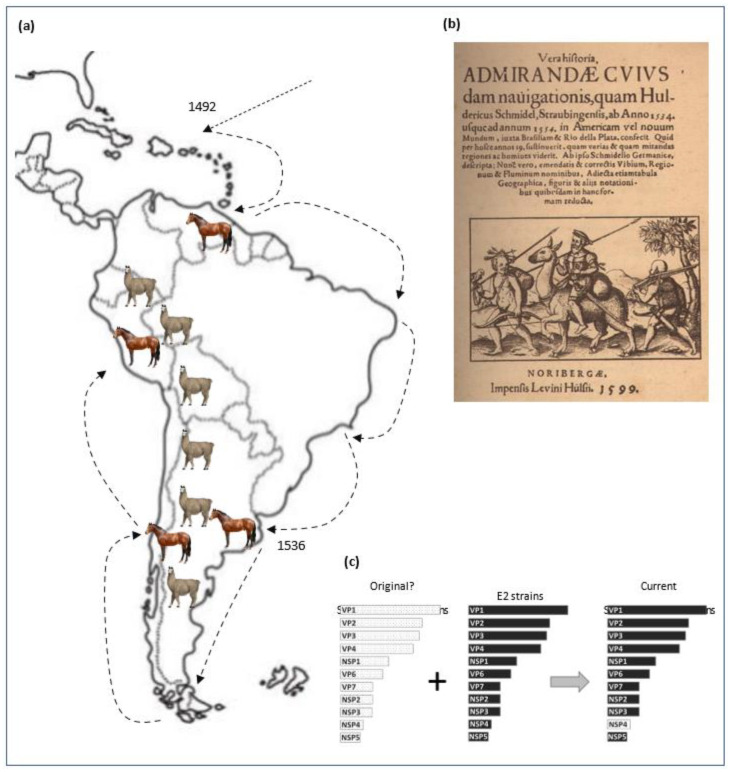
(**a**) The map shows the route followed by the Spanish including Pedro de Mendoza since the conquest of America. The numbers indicate the date of the foundation of the city (Buenos Aires: 1536; Santiago de Chile: 1541). The places where cows and horses were released are indicated by figures of horses. The territory occupied by wild guanacos is indicated with small figures of guanacos. (**b**) The diary of Mr. Schmidel, survivor of the expedition of 1534, showed Pedro de Mendoza riding a guanaco. Considering that Pedro de Mendoza died shortly after the expedition began, one could propose that he encountered the guanacos in Buenos Aires. (**c**) Hypothesis of the possible origin of the E12 gene is explained as a gene substitution event.

**Table 1 viruses-14-02506-t001:** Number of NSP4 E12 strains reported in South America.

Host	Country			
	Argentina	Brazil	Paraguay	Uruguay
Humans	-	1 ^a^	1 ^b^	-
Cows	30 *	-	-	10 ^c^
Horses	49 *	2 ^d^	-	-
Guanacos	2 ^e^	-	-	-
Goats	3 *^,f^	-	-	-

* Sequence strains reported by the authors in this work or previously. ^a^ Volotão et al., 2006. ^b^ Martinez et al., 2014. ^c^ Castells et al., 2019. ^d^ Volotão, E.M. and Gomez, M.M. Unpublished. ^e^ Matthijnssens et al., 2009. ^f^ Louge Uriarte et al., 2014.

## Data Availability

All sequences showed in Table 1 came from Badaracco A., Louge Uriarte E., and Miño S. thesis works, accession numbers OP676137-OP676207.

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
