# Peer review of "Evolution of Animal South American RVA Told by the NSP4 Gene E12 Genotype"

_viruses, 2022, doi:10.3390/v14112506_

Round 1

Reviewer 1 Report

The manuscript describes the evolution of the rotavirus NSP4 gene E12 genotype of rotavirus A. This genotype appears to be specific to the South American continent. The authors confirm the South American origin of that genotype and further show that its natural ancestral host was the Guanaco, an indigenous camelid species. In addition, the TMRCA is computed alongside the virus population dynamics. Interestingly, this shows a remarkable concordance with the introduction of horses and cows in South America by Europeans in the 16th century. Overall, the analysis is well described and appears sound. I only have minor comments that should be addressed:

-       Could you more clearly justify the choice of NSP4? Is it the sole genome segment that shows a genotype specific to the South American continent? What about other genome segments?

-       On line 88, could you specify what is meant by "not enough covering"?

-       Last references are lacking (from ref #30).

Author Response

Reviewer 1

The manuscript describes the evolution of the rotavirus NSP4 gene E12 genotype of rotavirus A. This genotype appears to be specific to the South American continent. The authors confirm the South American origin of that genotype and further show that its natural ancestral host was the Guanaco, an indigenous camelid species. In addition, the TMRCA is computed alongside the virus population dynamics. Interestingly, this shows a remarkable concordance with the introduction of horses and cows in South America by Europeans in the 16th century. Overall, the analysis is well described and appears sound. I only have minor comments that should be addressed:

  1. Could you more clearly justify the choice of NSP4? Is it the sole genome segment that shows a genotype specific to the South American continent? What about other genome segments?

During the last 20 years our group has been studying the epidemiology and pathogenicity of rotaviruses in several animal species resulting in several publications, including those of Matthijnssens et al, 2009 (https://doi.org/10.1128/jvi.02246-08), Garaicoechea et al, 2011 (https://doi.org/10.1016/j.vetmic.2010.08.032), Louge Uriarte et al, 2014 (https://doi.org/10.1016/j.vetmic.2014.03.013), Miño et al, 2017 (https://doi.org/10.1016/j.jevs.2017.09.008) and others. Our findings have revealed that rotavirus strains from different animal species always carried the NSP4 E12 genotype, while sharing the typical backbone. The other 10 genes possess genotypes that have been reported around the world and they do not show a geographic pattern like the NSP4 E12 genotype does.

For many years, we did not find an explanation for this observation until now when we performed a Bayesian analysis that allowed us to determine a plausible date, rather remarkably, with the date of the introduction of horses and cows in South America by Europeans in the 16th century. Therefore, we initiated a detailed investigation in an attempt to understand the history of cattle and wild animals in South America. At the same time, additional investigations were being carried out, all of which confirmed our hypothesis. Initially, we did not consider finding a geographic marker in an ordinary find. However, review of the historical literature provided additional that support to our hypothesis, which precipitated us to perform a comprehensive analysis of the current investigation as a standalone because of the importance it deserves. Hence, the current article with the hypothesis that we present here. This summary has been added to the manuscript (page 7, lines 1 to 18).

  1. On line 88, could you specify what is meant by "not enough covering"?

As requested by the reviewer, we now defined the 95% of the open reading frame as the cut-off to include a strain sequence into the analyses. All strain sequences under cut-off were excluded.

  1. Last references are lacking (from ref #30).

The references have been updated accordingly.

Reviewer 2 Report

General comments:

The manuscript describes the evolutionary relationships of RVAs carrying the E12 genotype detected in South America. The authors studied the evolution of these viruses in different hosts including humans, cattle, horses, goats, and camelids, by analyzing the tenth genomic segment of RVA which encodes the NSP4 enterotoxin. The NSP4 analysis of strains belonging to the E12 genotype, only detected in South America, showed that these branches are a monophyletic group, and it seems to have evolved from the interspecies transmission between South American camelids of farmed animals from Europe. Based on these observations, the authors suggest that the E12 genotype could be the genetic geographic marker in the RVA circulating in different animal species in South America. The methodology used is appropriate, the results obtained are robust and support the conclusions. Therefore, I recommend publishing the article.

Minor comments:

1. According to ICTV the spelling of family, genus and species should be in italics

2. RVA group A (RVA): the correct taxonomic term is species and not group

3. Throughout the text rotavirus does not need to be written in capital letters.

4. In the text, 40 bibliographic references are cited, but only 29 are listed in the references section.

Author Response

Reviewer 2

The manuscript describes the evolutionary relationships of RVAs carrying the E12 genotype detected in South America. The authors studied the evolution of these viruses in different hosts including humans, cattle, horses, goats, and camelids, by analyzing the tenth genomic segment of RVA which encodes the NSP4 enterotoxin. The NSP4 analysis of strains belonging to the E12 genotype, only detected in South America, showed that these branches are a monophyletic group, and it seems to have evolved from the interspecies transmission between South American camelids of farmed animals from Europe. Based on these observations, the authors suggest that the E12 genotype could be the genetic geographic marker in the RVA circulating in different animal species in South America. The methodology used is appropriate, the results obtained are robust and support the conclusions. Therefore, I recommend publishing the article.

Minor comments:

  1. According to ICTV the spelling of family, genus and species should be in italics

As requested by the reviewer, the spelling was corrected.

  1. RVA group A (RVA): the correct taxonomic term is species and not group

As requested by the reviewer, the term was corrected.

  1. Throughout the text rotavirus does not need to be written in capital letters.

As requested by the reviewer, rotavirus is no longer in capital letters.

  1. In the text, 40 bibliographic references are cited, but only 29 are listed in the references section.

As requested by the reviewer, the references have been updated.
